# Effect of the Synthesis Conditions on the Morphology, Luminescence and Scintillation Properties of a New Light Scintillation Material Li$_2$CaSiO$_4$:Eu$^{2+}$ for Neutron and Charged Particle Detection

**Ilia Komendo** [1,2,*] , **Vitaly Mechinsky** [2], **Andrei Fedorov** [2], **Georgy Dosovitskiy** [1], **Victor Schukin** [1], **Daria Kuznetsova** [1,2], **Marina Zykova** [3], **Yury Velikodny** [4] and **Mikhail Korjik** [1,2]

1  NRC "Kurchatov Institute"—IREA, Bogorodskiy val 3, 107076 Moscow, Russia
2  NRC "Kurchatov Institute", Kurchatov Square 1, 123182 Moscow, Russia
3  Department of Chemistry and Technology of Crystals, Moscow University of Chemical Technology of Russia, Miusskaya 9, 125047 Moscow, Russia
4  Department of Chemistry, Moscow State University, Kolmogorova 1-3, 119991 Moscow, Russia
*  Correspondence: ilia.komendo@cern.ch; Tel.: +7-9168312556

**Abstract:** In the present article, the influence of the activator concentration and impurity content of raw materials on the luminescence and scintillation properties of Li$_2$CaSiO$_4$ was studied. Polycrystalline powder material was obtained by the sol–gel method. It was shown that europium had limited solubility in the host lattice with a limiting concentration proximate to 0.014 formula units. The maximum intensity of photoluminescence was observed with a divalent europium concentration of 0.002 formula units; the light yield under alpha-particle excitation was measured to be 21,600 photons/MeV for ~200 μm of coating, and under neutron excitation, it was calculated to be 103,800 photons/n, the scintillation kinetics was characterized by an effective decay time of 157 ns. These properties and the transparency in the visible spectrum make it possible to produce scintillation screens with a coating of Li$_2$CaSiO$_4$ for detecting neutrons, alpha particles and low-energy beta radiation. The low Z$_{eff}$ (~15) of this compound makes it less sensitive to gamma rays. The 480 nm blue emission peak makes this material compatible with most commercial PMT photocathodes, CCD cameras and silicon photomultipliers, which have a maximum quantum efficiency in the blue–green spectral region.

**Keywords:** phosphor; lithium calcium silicate; luminescence; scintillator; europium; neutron detection

## 1. Introduction

The scintillation method for detecting ionizing radiation is used in many areas of science and technology, for example, in research facilities, inspection and medical equipment, etc. For different types of radiation, different scintillators are used. Lithium- and gadolinium-containing compounds are widely used for neutron detection due to the large capture cross-section for thermal and epithermal neutrons in the nuclei of these elements [1]. In the case of lithium, neutron registration occurs due to the reaction with the nucleus of the lithium-6 atom with the emission of an alpha particle and a tritium nucleus with a total energy of 4.8 MeV (Equation (1)), their absorption in the scintillator material and subsequent emission of visible light photons.

$$^6Li + n \rightarrow {}^4He + {}^3H + 4.8 \ MeV \tag{1}$$

The interaction of a neutron with the nucleus of a gadolinium atom is followed by the emission of a number of gamma quanta and conversion electrons (Equation (2)) with a

total energy of ~8 MeV, and the number of emitted gamma quanta depends on the neutron energy. In this case, the scintillator is excited predominantly by conversion electrons.

$$^{n}Gd + n \rightarrow {}^{n+1}Gd + \gamma + e^{-} \tag{2}$$

Gadolinium-containing materials are commercially available, common examples are $Gd_2O_2S$:Tb (GOS), which is used as a powder coating in neutron and gamma radiography equipment or as ceramics in screening and tomography equipment [2–4]. Another material is $Gd_3Al_2Ga_3O_{12}$:Ce (GAGG) or related compounds with a garnet structure, which are applied as single crystals or composites in various types of neutron detectors [5–8]. Despite widespread application and commercial availability, gadolinium-based scintillators have a significant drawback regarding the close efficiency of registering target and background events. Therefore, methods of discrimination of secondary and background gamma quanta are widely used in the neutron experiment techniques [9].

It is known that a higher nuclear charge provides more effective gamma ray absorption within a substance; therefore, light lithium-containing scintillators became the objects of interest. Moreover, the products of the reaction between lithium and neutron atoms are alpha particles and the nuclei of tritium atoms, which have short path lengths even in low-$Z_{eff}$ substances. Well-known examples of lithium-containing scintillators are $^6$LiI:Eu, $Cs_2LiLaBr_6$, $CsLiYCl_6$ (CLYC) and $Li_3YCl_6$ [10–12]. However, these compounds have an effective charge $Z_{eff}$ of ~45 for the first two compounds and ~30 for the last ones. Among the variety of scintillators, a mixture of $^6$LiF, which plays the role of a radiation converter, and ZnS(Ag) scintillator stands out due to a relatively low effective charge and high light yield. The disadvantage of this composite is its opacity toward scintillation photons; therefore, despite the high light yield under alpha particles, the efficiency of light extraction from the LiF/ZnS layers is low. In addition, the decay kinetics of ZnS:Ag scintillations equals 3–6 μs, which limits the use of this scintillator for detecting intense fluxes of neutrons and other charged particles.

To date, the attention of researchers is focused on the search for new light scintillators for detecting neutrons that have a high light yield and fast decay kinetics. Recently, crystals of the eutectics $Li_2SrCl_4$/$LiSr_2Cl_4$ and $LiCl$/$BaCl_2$ [13,14] were grown. In such crystals, the alkaline earth metal atoms (Ca, Ba) are partially substituted by an activator—divalent europium. In our work, we chose a lighter matrix for this activator, $Li_2CaSiO_4$, in which calcium ions are isovalently substituted by europium. The $Li_2O$–$SiO_2$–$CaO$ system has been actively studied over the past decade as a potential blue phosphor for white LEDs or phosphors with variable chromaticity [15–18]. However, the scintillation properties of the material activated with europium ions have not yet been studied. $Li_2CaSiO_4$ has a low $Z_{eff}$ of ~15, which means that one should expect a lower sensitivity to background gamma quanta. In addition, it is possible to use raw materials enriched with the isotopes of lithium-6 or lithium-7, which can influence the sensitivity of this scintillator to neutrons or charged particles. The silicate matrix makes it possible to produce glass from such or related compositions [19,20] and grow single crystals [21], which potentially may significantly expand the application area of $Li_2CaSiO_4$:Eu.

## 2. Results and Discussion

### 2.1. Synthesis and Characterization

Samples of polycrystalline powders with a composition of $Li_2Ca_{1-x}Eu_xSiO_4$ were prepared, with X = 0.0006, 0.0023, 0.0050, 0.0132 and 0.0329 formula units. The synthesis procedure is described in Section 3.8. The powders consisted of particles with irregular morphology, images from an optical microscope and a scanning electron microscope are presented in Figure 1a,b, respectively. It can be seen that the particles are transparent in the visible range (Figure 1a). This is noteworthy from the viewpoint of potential applications of $Li_2CaSiO_4$ as coatings—the transparency of the particles will allow better light extraction from a thick scintillator layer compared to the widely used opaque LiF/ZnS composite. The presence of lithium ions in the composition of the compound makes it possible to create

two-component screens using a transparent binder. Panel 1c shows a 100 µm $Li_2CaSiO_4$ powder layer on the quartz substrate.

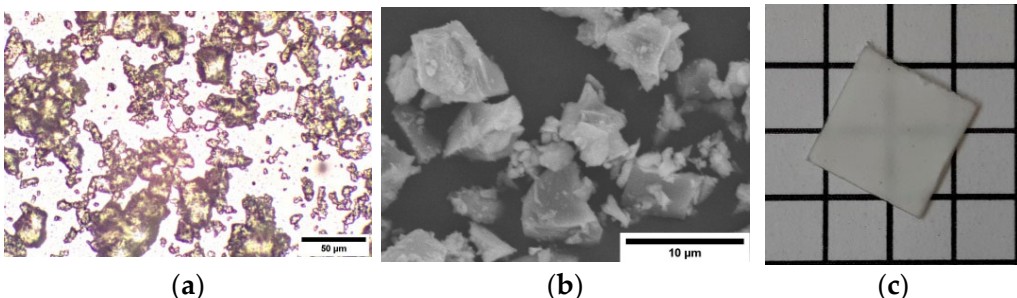

**Figure 1.** Images of $Li_2CaSiO_4$:Eu particles obtained with an optical microscope (**a**), scanning electron microscope (**b**) and a photograph of 100 µm layer on quartz substrate (**c**).

The phase composition of the samples was studied by X-ray powder diffraction. The diffraction patterns of samples are shown in Figure 2. The samples contain traces of $Li_2SiO_3$ and $Li_3Si_2O_7$ in the sample with x = 0.0329 (see below), which did not affect the results of measurements of the luminescence properties.

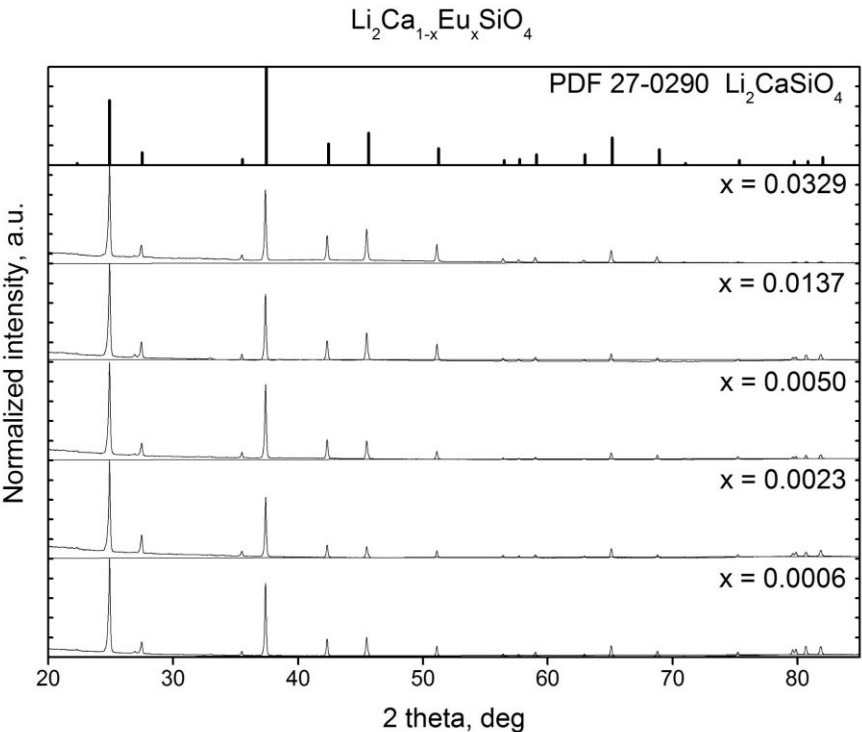

**Figure 2.** Powder X-ray diffraction pattern of $Li_2Ca_{1-x}Eu_xSiO_4$ samples.

Divalent europium has an ionic radius close to calcium in octahedral coordination equal to 1.25 Å (1.12 Å for $Ca^{2+}$); therefore, it substitutes it in the crystal lattice of lithium calcium silicate. However, with an increase in the concentration of europium in the matrix, an increase in the volume of the unit cell is observed as seen in Figure 3a. Calculated unit cell a, c parameters and volumes for different europium concentration are shown in Table 1.

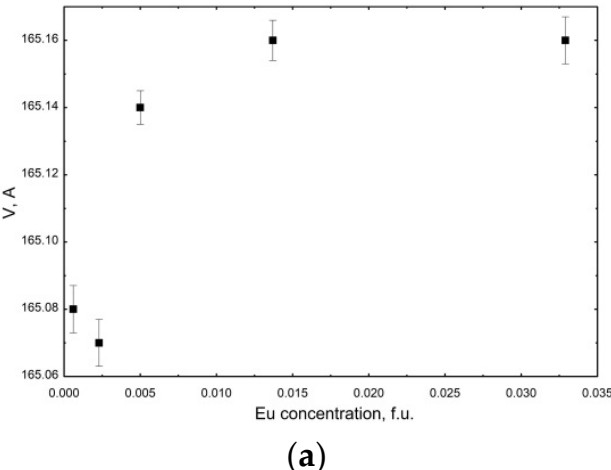
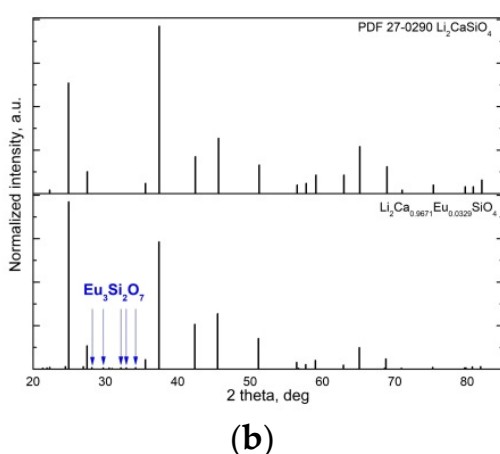

**(a)** **(b)**

**Figure 3.** Dependence of the unit cell volume of $Li_2Ca_{1-x}Eu_xSiO_4$ on the europium concentration (**a**). Diffraction pattern of a sample with a concentration of europium 0.0329 f.u. (**b**).

**Table 1.** Calculated unit cell parameters of $Li_2Ca_{1-x}Eu_xSiO_4$.

| Eu Concentration, form. u. | a, Å | c, Å | v, Å |
|---|---|---|---|
| 0.0006 | 5.0467(8) | 6.4813(15) | 165.08(7) |
| 0.0023 | 5.0464(9) | 6.482(14) | 165.07(7) |
| 0.0050 | 5.0472(6) | 6.4826(9) | 165.14(5) |
| 0.0137 | 5.0472(7) | 6.4834(12) | 165.16(6) |
| 0.0329 | 5.0463(8) | 6.4859(15) | 165.16(7) |

Apparently, europium has limited solubility in the crystal host lattice, since the volume of the unit cell reaches a plateau at a concentration of europium of 0.0137 formula units. Thus, the concentration of europium ~0.012–0.014 formula units might be the limit for this matrix. This can be supported by the X-ray diffraction pattern of the sample with 0.0329 formula units of europium, which shows traces of the presence of europium in the form of an independent phase of $Eu_3Si_2O_7$ (Figure 3b). The same was found in [17].

### 2.2. Luminescent Properties

The photoluminescence spectrum of $Li_2CaSiO_4$:$Eu^{2+}$ has one broad band at 480 nm (Figure 4), which corresponds to the interconfigurational radiative transition in $Eu^{2+}$ ions from the lower level of the $4f^65d^1$ configuration to the ground state of the $4f^7$ configuration $^8S_{7/2}$-state [15,22]. The 480 nm luminescence band is excited in two broad bands, peaked at 289 and 376 nm. With an increase of the europium concentration, a slight shift of the luminescence peak to the long wavelength region is observed. The FWHM of the peaks remains constant up to a europium concentration of 0.0023 formula units and is equal to 31.9 nm (0.15 eV). As the activator concentration increases, FWHM decreases to 29.8 nm at a concentration of 0.0329 f.u. Upon excitation at a wavelength of 395 nm [23], which corresponds to the excitation of $Eu^{3+}$ luminescence, low-intensity luminescence bands in the red region of the spectrum are also observed (see inset to Figure 4). In the dependence of the photoluminescence intensity on the activator concentration, a maximum is observed in the region of 0.002 formula units (Figure 5).

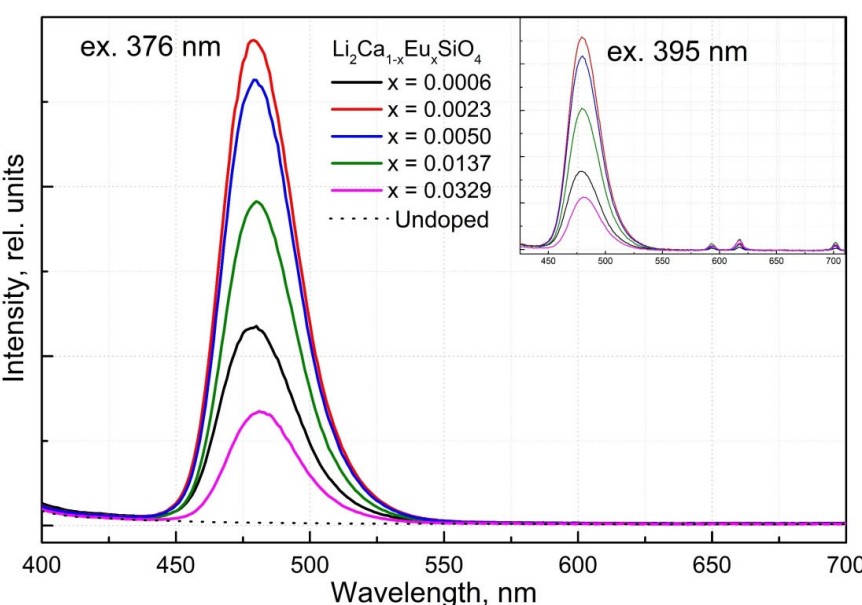

**Figure 4.** Photoluminescence spectra of $Li_2Ca_{1-x}Eu_xSiO_4$ with different activator concentrations. In the inset, similar spectra are shown but were obtained at an excitation wavelength of 395 nm.

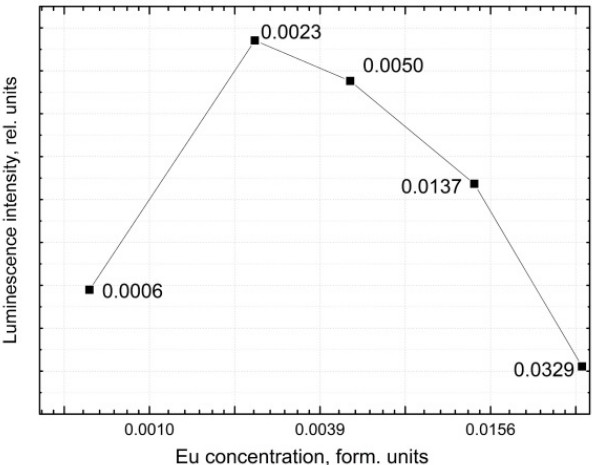

**Figure 5.** Dependence of $Li_2Ca_{1-x}Eu_xSiO_4$ photoluminescence intensity at excitation 376 nm on Eu concentration.

This graph shows that luminescence quenching is predominantly concentration related. This conclusion is confirmed by measurements of the photoluminescence kinetics.

The photoluminescence decay curves measured at room temperature are shown in Figure 6.

Approximation of the luminescence kinetics was carried out by a set of exponential functions. The results of the approximation are shown in Table 2. At the lowest concentration of activator, the kinetics is close to a single exponential with a decay time of a half microsecond. Further increases in the doping concentration provide an acceleration of the initial part of the kinetics due to concentration quenching. It results in the appearance, at approximation, of the shorter components. At a multicomponent approximation of the luminescence kinetics, the drop in the luminescence yield can be characterized by a change in the effective decay constant.

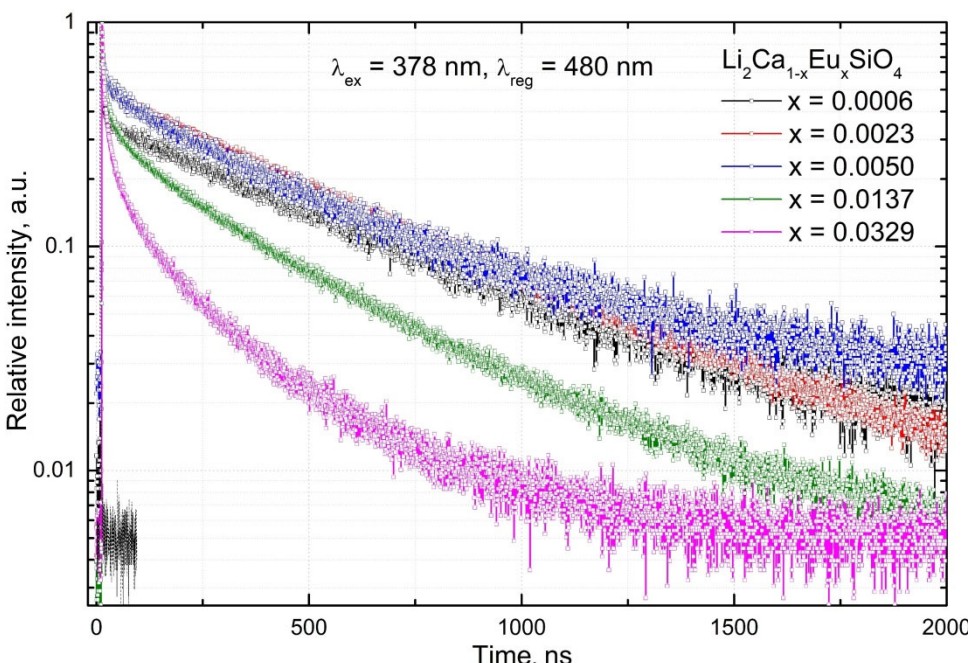

**Figure 6.** Photoluminescence decay kinetics of $Li_2Ca_{1-x}Eu_xSiO_4$ samples measured at room temperature. The excitation pulse (instrument response function) is shown by a dashed black line.

**Table 2.** Photoluminescence decay parameters of $Li_2Ca_{1-x}Eu_xSiO_4$.

| [Eu], f.u. | $\tau$, ns | *p*, % | $\tau$ * Effective, ns |
|---|---|---|---|
| 0.0006 | 512.7 | 100 | 512.7 |
| 0.0023 | 89.0<br>514.0 | 2<br>98 | 473.7 |
| 0.0050 | 482.7<br>91.5 | 94<br>6 | 393.8 |
| 0.0137 | 458.1<br>167.0<br>22.1 | 78<br>20<br>2.00 | 262.1 |
| 0.0329 | 352.7<br>122.9<br>24.3 | 56<br>38<br>6 | 138.0 |

* An effective decay time is evaluated as $\sum_i \tau_i f_i$, where $f_i$ is a fraction of the component $\tau_i$.

The effective decay time decreases with increasing europium concentration in proportion to the decrease in the integrated luminescence intensity, $\tau$.

### 2.3. Scintillation Properties

Scintillation measurements were carried out with the samples of coatings on aluminum substrates with dimensions of 15 × 15 mm. The composition of the coating was $Li_2Ca_{1-x}Eu_xSiO_4$ powder sieved through a 150-mesh sieve and a polyacrylic binder in a volume ratio of 95/5. The layer density was ~35 mg $Li_2Ca_{1-x}Eu_xSiO_4/cm^2$, and the layer thickness was ~200 μm. The samples were fixed in a light-tight volume at an angle of 45° to a [241]Am source of alpha-particles with an energy of 5.5 MeV; scintillation photons were recorded on a PMT Hamamatsu R329. A YAG:Ce single crystal with a known light yield of 6000 ph./MeV at alpha-particle registration was used as a reference sample for evaluating the light yield. Pulse height spectra measured at room temperature are shown in Figure 7.

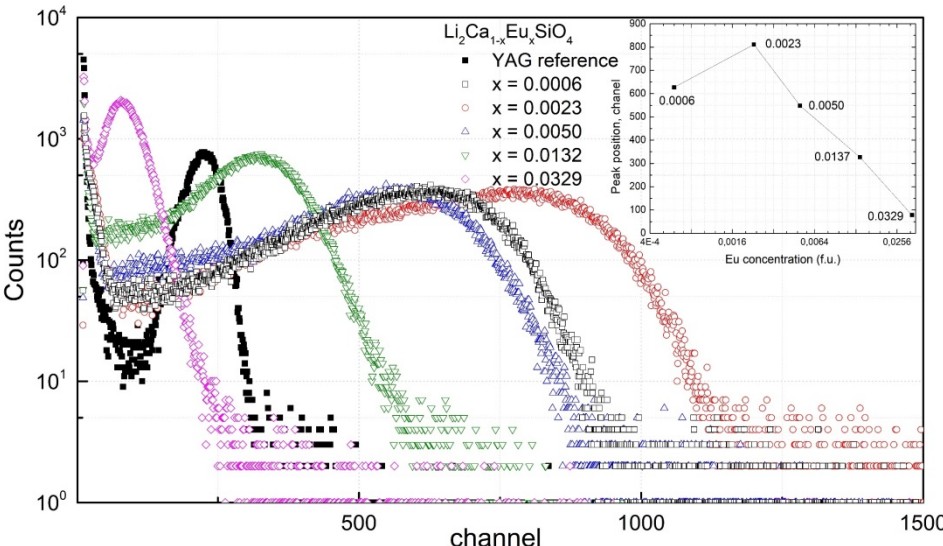

**Figure 7.** Pulse height spectra of 5.5 MeV alpha particles of the $^{241}$Am source recorded with $Li_2Ca_{1-x}Eu_xSiO_4$ samples.

The inset of Figure 7 shows the dependence of the position of the total absorption peak maximum on the activator concentration. The dependence is close to the integrated photoluminescence intensity and has a maximum at a europium concentration of ~0.002 formula units. According to the peak position, the light yield can be estimated relative to the reference sample—YAG:Ce using Equation (3):

$$I_{sample} = I_{ref} \frac{L_{sample}}{L_{ref}}, \tag{3}$$

where $I_{sample}$—light yield of the sample;

$I_{ref}$—known light yield of the reference (YAG:Ce = 6000 ph./MeV);

$L_{sample}$—photoabsorption peak center position of the sample;

$L_{ref}$—photoabsorption peak center position of the reference.

One can calculate the values of the light output under 5.5 MeV alpha-particles as well as for thermal neutrons, taking into account the total energy release (alpha + triton) according to the Equation (1) (Table 3).

**Table 3.** Light yield of $Li_2Ca_{1-x}Eu_xSiO_4$ samples under alpha-particle excitation and calculated one of thermal neutron excitation. Error of light yield measurements was estimated to be ±1000 ph./MeV.

| [Eu], f.u. | Peak Center Position, Channel | Light Yield, ph./MeV | Light Yield, ph./n |
|---|---|---|---|
| 0.0006 | 627 | 16,700 | 80,200 |
| 0.0023 | 811 | 21,600 | 103,600 |
| 0.0050 | 548 | 14,600 | 70,100 |
| 0.0137 | 327 | 8700 | 41,800 |
| 0.0329 | 78 | 2080 | 10,000 |
| YAG:Ce | 225 | 6000 | - |

Optimization of the activator concentration makes it possible to reach a light yield of ~104,000 photon/MeV under thermal neutron excitation.

Scintillation kinetics was measured by the start–stop method with a $^{22}$Na source. The scintillations decay curves are shown in Figure 8.

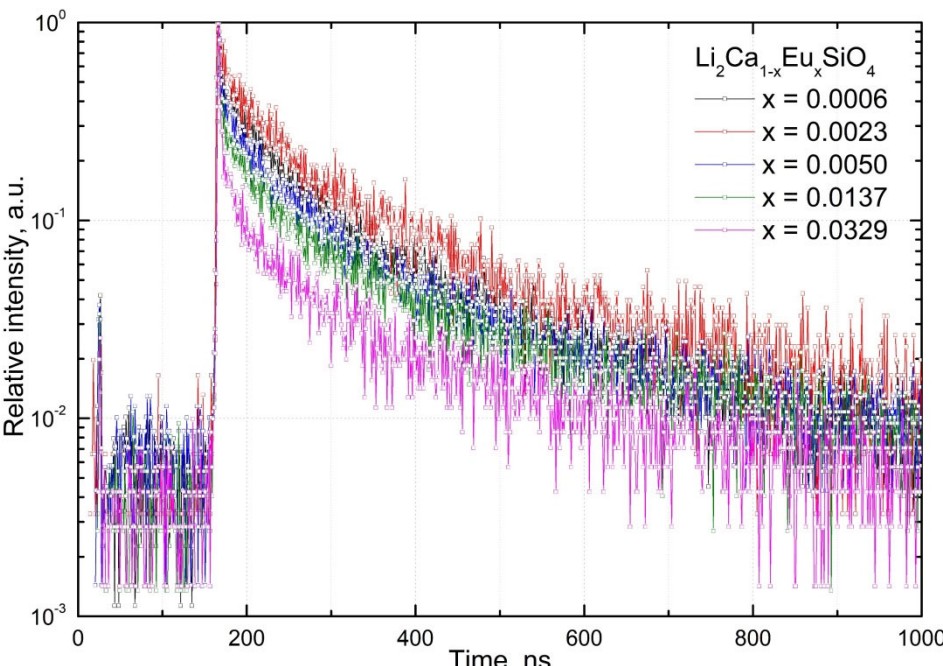

**Figure 8.** Normalized scintillation decay kinetics of $Li_2Ca_{1-x}Eu_xSiO_4$.

The distribution of scintillation constants is given in Table 4.

**Table 4.** Scintillation decay parameters of $Li_2Ca_{1-x}Eu_xSiO_4$.

| [Eu], f.u. | $\tau$, ns | $p$, % | $\tau$ Effective, ns |
|---|---|---|---|
| 0.0006 | 40.7<br>130.1<br>660.4 | 22.89<br>39.84<br>37.27 | 108.1 |
| 0.0023 | 48.7<br>192.9<br>1729.3 | 20.39<br>37.67<br>41.94 | 156.7 |
| 0.0050 | 38.6<br>148<br>1499.6 | 16.18<br>35.95<br>47.87 | 144.1 |
| 0.0137 | 23.6<br>103.3<br>631.7 | 13.98<br>34.02<br>52.00 | 99.6 |
| 0.0329 | 19.1<br>106<br>932 | 14.10<br>24.30<br>61.60 | 96.8 |

Similar to the case of photoluminescence kinetics, the effective scintillation kinetics decay time tends to decrease with increasing activator concentration. It should be mentioned that compared to the lithium-containing $Eu^{2+}$ scintillators, $LiCa_2I_5$ and $LiCa_2Br_5$, $Li_2CaSiO_4$ has a scintillation decay time of 156.7 ns contrary to 1.2–1.4 µs for the mentioned halide compounds [24].

### 2.4. Impurities Effect

In order to study the effect of raw material purity on the luminescent and scintillation properties of $Li_2CaSiO_4$:Eu, we purified the reagent-grade $Li_2CO_3$ and $CaCO_3$ to purity of 99.984% and 99.992%, respectively. Tetraethoxysilane did not require further purification. Figure 9 shows the pulse height spectra of samples obtained from commercial reagents

and from purified ones. The inset in Figure 9 shows the photoluminescence spectra for the same samples.

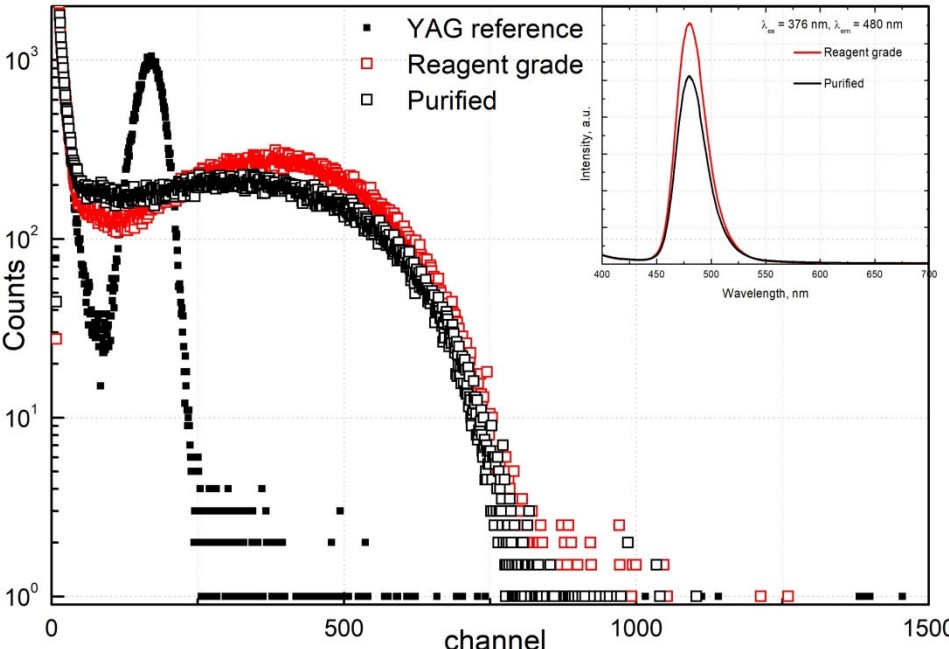

**Figure 9.** Pulse height spectra of 5.5 MeV alpha particles of the $^{241}$Am source recorded for $Li_2Ca_{1-x}Eu_xSiO_4$ samples obtained from raw materials of different purity.

The concentration of the activator for all samples was the same and equaled 0.0050 formula units. From the above calculation, it can be estimated that the light output for samples is nearly within the error of measurements: 13,400 ± 1000 vs. 11,500 ± 1000 ph./MeV. On the other hand, variations of the intensities of photoluminescence were found to be more pronounced. Samples from commercial reagents had ~30% higher photoluminescence intensity. The effect obtained can be explained by the sensitizing effect of impurities: there are studies in which an increase in the intensity of photoluminescence of $Li_2CaSiO_4:Eu^{2+}$ was observed upon the addition of codopants [25,26]. It is notable that a random set of impurities can give a similar effect as a specially chosen element, although the quenching effect of impurities will be inferior to their positive contribution to the luminescence intensity.

## 3. Materials and Methods

### 3.1. Reagents

We used reagents of analytical quality with additional purification, in particular, $CaCO_3$, 99.313% purity; $Li_2CO_3$, 99.934% purity; tetraethoxysilane, $Eu_2O_3$ 5N; and deionized water with a resistivity ≥18 MΩ/cm. To study the effect of impurities on the properties of $Li_2CaSiO_4$, the $CaCO_3$ and $Li_2CO_3$ raw materials were additionally purified to a purity of 99.992% and 99.984%, respectively.

The impurity composition of the raw materials, as well as the intermediate products, was measured by inductively coupled plasma atomic emission spectrometry (ICP-MS).

### 3.2. Inductively Coupled Plasma Mass Spectrometry (ICP-MS)

Samples (250–300 mg) were dissolved in 5 mL of nitric acid ($HNO_3$) (7N) purified by a surface distillation system (BSB-929-IR, Berghof, Germany) in a polypropylene (PP) test tube. The deionized water was produced by an Aqua- MAX-Ultra 370 Series setup (Young Lin Instruments, Seoul, Korea) and had 18.2 MΩ cm electrical resistance and 99.99999 wt.% purity. A 1 mL aliquot taken from the resultant solution was transferred to a PP test tube, and then, the solution was brought to 10 mL by adding water. The solution

thus prepared was analyzed by inductively coupled plasma mass spectrometry (ICP-MS). The measurements were performed using a NexION 300D ICP-MS system (PerkinElmer, Waltham, USA) in collision mode (kinetic energy discrimination, KED) [27].

### 3.3. Photoluminescence Studies

Photoluminescence spectra were measured using a Fluorat-Panorama-2M spectrometer (Lumex, St. Petersburg, Russia) with a high-pressure xenon lamp as the excitation source.

Photoluminescence kinetics were measured on a Pico Quant 250 spectrofluorometer (Pico Quant, Berlin, Germany) with pulse LED excitation having a pulse width of 500 ps. A PLS-370 excitation source was used with the maximum excitation intensity at a wavelength of 378 nm.

### 3.4. Powder X-ray Diffraction

The phase composition of the powder samples was studied using a Bruker D2 Phaser diffractometer (Bruker, Billerica, USA) with a LYNXEYE XE-T detector. Cu k$\alpha$ radiation was used, with a 2-theta range of 20–90° in steps of 0.01 degrees, integration time 0.3 s.

### 3.5. Scintillation Measurements

The evaluation of the light yield with the gamma quanta of the transparent single crystalline samples is dependent on shape and surface quality [28]. In our study, we used low-range penetrating particles for scintillation excitation. Pulse height spectra of alpha particles from a 241Am source were recorded in 45° geometry according to the technique described in [29] with a Hamamatsu R329 photomultiplier tube and ORTEC electronics; the spectrum acquisition time was 300 s. Samples of $Li_2Ca_{1-x}EuxSiO_4$ coatings on aluminum substrate were measured, and a YAG:Ce single crystal with a ground surface was used as the reference. The light yield of the reference was 6000 ph./MeV under alpha-particle excitation. The penetration depth of 5 MeV alpha particles in the $Li_2CaSiO_4$ was estimated by GEANT4 simulation to be ~6 μm. Therefore, one can ignore the influence of the shapes of the samples and the reference on the results of light yield measurements. The difference in the quantum efficiency of the PMT photocathode at the YAG:Ce and Li2CaSiO4:Eu emission wavelengths (9% at 540 nm and 16% at 480 nm, respectively) was taken into account at evaluation of the light yield.

The test bench included a $^{22}$Na source of annihilation γ-quanta (511 keV) and two measurement channels: a start channel, including a $CaF_2$ scintillation crystal connected to a Photonis XP2020Q photomultiplier (PMT), and a stop channel with a Photonis XP2020 PMT (Both are from Photonis, Lancaster, USA). Response functions were measured to be $1.2 \pm 0.2$ ns. ORTEC electronic modules were used to amplify and process the signals.

Scanning electron microscope images were obtained on Hitachi SU1510 (Tokyo, Japan) and Altami 5M (Altami, St. Petersburg, Russia) instruments, respectively.

### 3.6. Additional Purification of $CaCO_3$ and $Li_2CO_3$

Calcium carbonate was purified by conversion to formate [30] by interacting with a solution of formic acid (Equation (4)). The calcium formate was crystallized, washed and recrystallized twice. Then, it was converted back to carbonate by interaction with a $(NH_4)_2CO_3$ solution (Equation (5)).

$$CaCO_3 + 2HCOOH \rightarrow Ca(COOH)_2 + H_2O + CO_2 \tag{4}$$

$$Ca(COOH)_2 + NH_4HCO_3 + NH_3 \cdot H_2O \rightarrow CaCO_3 \downarrow + 2NH_4COOH \tag{5}$$

An aqueous suspension was prepared from lithium carbonate, through which carbon dioxide was bubbled, with the formation soluble hydrogen carbonate (Equation (6)).

The resulting solution was filtered and heated; when heated, purified lithium carbonate precipitated (Equation (7)).

$$Li_2CO_3 + H_2O + 2CO_2 \rightarrow 2LiHCO_3 \tag{6}$$

$$2LiHCO_3 \rightarrow Li_2CO_3 \downarrow + 2CO_2 + H_2O \tag{7}$$

The detailed impurity composition is specified in Supplementary Materials Figures S1 and S2.

### 3.7. Synthesis of CaCO₃:Eu

Calcium carbonate and europium oxide were dissolved in dilute nitric acid and then mixed with a solution of $(NH_4)_2CO_3$. The resulting $CaCO_3$:Eu precipitate was washed and dried to constant weight in an oven. After that, the concentrations of Ca and Eu in the residue were determined by ICP-MS. The measurement results are shown in Table 5.

**Table 5.** Nominal concentrations of europium in $Ca_{1-x}Eu_xCO_3$ and actually determined by ICP-MS.

| Nominal Eu Concentration, Form. u. | [Ca] Measured, % Mass | [Eu] Measured, % Mass | Real Eu Concentration, Form. u. |
|---|---|---|---|
| 0.0005 | 37.62 | 0.082 | 0.0006 |
| 0.002 | 31.03 | 0.274 | 0.0023 |
| 0.005 | 39.02 | 0.723 | 0.0050 |
| 0.008 | 28.72 | 1.515 | 0.0137 |
| 0.032 | 32.26 | 4.158 | 0.0329 |

Differences between the nominal and real concentrations of the activator are apparently caused by the partial underprecipitation of calcium carbonate.

### 3.8. Synthesis of Li₂CaSiO₄:Eu²⁺

Synthesis of $Li_2Ca_{1-x}Eu_xSiO_4$ was carried out as follows: a stoichiometric amount of water necessary for hydrolysis was added to tetraethoxysilane, and a few drops of dilute nitric acid were added to initiate hydrolysis. After hydrolysis was finished, stoichiometric amounts of $Ca_{1-x}Eu_xCO_3$ obtained according to the procedure described above, where x = 0.0006, 0.0023, 0.0050, 0.0137 and 0.0329, and $Li_2CO_3$ were added to the hydrolyzed tetraethoxysilane. The suspension was left on a magnetic stirrer to homogenize the components. Then, the suspension formed a homogeneous elastic gel-like mass, which was dried at 100 °C, crushed in an agate mortar and sieved through a 150-mesh sieve. The resulting precursor was heat-treated at 850 °C, for at least 2 h to decompose carbonates and pre-synthesis the $Li_2Ca_{1-x}Eu_xSiO_4$ phase, after which the powder was again sieved through a 150-mesh sieve and sent to a tube furnace with an atmosphere of argon and hydrogen mixture (94% and 6%, respectively) at a temperature of 900 for 2 h to reduce europium to the valence state 2+ and for the final synthesis of $Li_2CaSiO_4$:Eu²⁺.

### 4. Conclusions

The dependence of the luminescent and scintillation properties of the $Li_2CaSiO_4$:Eu²⁺ compound on the concentration of the activator and the presence of impurities was studied. It is shown that the maximum intensity of photoluminescence and scintillation is observed at an activator concentration of 0.002 formula units. At the same time, europium has a limited solubility in the $Li_2CaSiO_4$ matrix, which can be approximately estimated at 0.014 formula units. The $Li_2Ca_{0.008}Eu_{0.002}SiO_4$ composition has a sufficiently high scintillation light yield of about 21,600 ph./MeV upon excitation by 5.5 MeV alpha-particles. Light yield under neutrons was estimated to be 103,000 ph./neutron. It is notably higher than that of well-known neutron scintillators such as cerium-doped $Cs_2LiYCl_6$, $Cs_2LiLuBr_6$ and $Rb_2LiLaBr_6$ which have the light yield of 70,000, 88,000 and 83,000 photon/neutron,

respectively [31,32]. Such properties make this material promising for use as layers for detecting various types of corpuscular radiation, including neutrons, when using raw materials enriched in the lithium-6 isotope.

**Supplementary Materials:** The following supporting information can be downloaded at: https://www.mdpi.com/article/10.3390/inorganics10090127/s1, Figure S1: Impurity content in reagent grade $CaCO_3$ (red columns) and purified (black columns); Figure S2: Impurity content in reagent grade $Li_2CO_3$ (red columns) and purified (black columns).

**Author Contributions:** Data curation, V.M.; funding acquisition, G.D. and D.K.; investigation, I.K., V.M., A.F., V.S., M.Z. and Y.V.; methodology, I.K.; project administration, M.K.; resources, V.M., A.F., M.Z. and Y.V.; supervision, G.D. and D.K.; writing—original draft, I.K.; writing—review and editing, M.K. All authors have read and agreed to the published version of the manuscript.

**Funding:** This research was funded by the NRC "Kurchatov institute" (No. 2076, dated 24 August 2021). Analytical studies have been carried out using the scientific equipment of CKP NRC "Kurchatov Institute—IREA", with the financial support of the project by the Russian Federation represented by the Ministry of Education and Science of Russia, Agreement No. 075-11-2021-070, dated 19 August 2021.

**Conflicts of Interest:** The authors declare no conflict of interest.

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
