# Peer review of "Effect of the Synthesis Conditions on the Morphology, Luminescence and Scintillation Properties of a New Light Scintillation Material Li2CaSiO4:Eu2+ for Neutron and Charged Particle Detection"

_inorganics, doi:10.3390/inorganics10090127_

Round 1

Reviewer 1 Report

The manuscript entitled “Effect of the synthesis conditions on the morphology, luminescence and scintillation properties of a new light scintillation material Li2CaSiO4:Eu2+ for neutrons and charged particles detection” by Ilia Komendo et al. is interesting however following points should be clarified.

Since scintillator must be chemically stable with high quantum yield and short decay time (lifetime) I would be interested in following:

1. Did authors investigate the effect of hydrolysis and presence of potential carbonates on measured properties (lifetime, infrared spectroscopy)?

2. The determination of lifetimes is not clear. Authors only stated that they used multiexponential function and 1 μs pulse width with 25 Hz repetition rate. I would be interesting how the nanosecond lifetimes (shorter than pulse width 1 μs) were extracted from decay data and what is the real accuracy of data fits (reduced Chi-square values) and lifetime values? Could authors provide instrument response function (pulse) and add it into data of Fig. 6? If the reconvolution has not been used, the obtained lifetimes are probably faulty since they are affected by pulse kinetics. Moreover, the decay data of Fig. 6 should be normalized between background (minimum) and maximum values, i.e. not only to maximum.

3. Please, add the instrument response function (pulse) into Fig. 8.

4. In Fig. 7 should be added background of measurement to proof that the observed spectra correspond only to samples emission. What is the reason of large signal distortion at longer channels? Has been Compton scattering taken into account?

5. I would be interested whether it is possible to compare pulse height spectra of samples of totally different geometry, i.e. single crystalline bulk YAG:Ce garnet to 200 μm thick composite (Li2CaSiO4:Eu/polyacrylate) film on Al substrate? For example, the different geometry studied in [M. Sasano, H. Nishioka, S. Okuyama et al., Nuclear Instruments and Methods in Physics Research A 715 (2013) 105] affects the resulting light yield. Therefore, the additional measurement of YAG:Ce prepared in same way like Li2CaSiO4:Eu samples, i.e. crushed and polyacrylate bonded to form a 200 μm thick film, is highly desired to be measured as a reference. It may affect the determination of the light yield by using the Eq. (3). Moreover, the sensitivity of detector to number of detected photons is not linear and detector saturation may be present at high-count rates. Did authors consider those facts in determination of light yield?

6. Authors stated that the studied samples may be utilized for scintillation. However, the studied samples have too long decay times thus, those materials could be hardly utilized as scintillators where is desired lifetime in scale of <100 ns (due to presence of Eu3+ traces?), rather approaching the 1–10 ns. Nevertheless, the present study is important but the above mentioned points must be carefully clarified.

Author Response

Dear Reviewer,

Thank you for a valuable comments! 

Please see the attachment our revised manuscript. All corrections were made with MS Word track changes function. 

Below please see our response to your comments.

Point 1: Did authors investigate the effect of hydrolysis and presence of potential carbonates on measured properties (lifetime, infrared spectroscopy)?

Response 1: Our XRD data confirm that after heat treatment (pre-synthesis) at 850 °С the Li2CaSiO4 phase is forming with an absence of residual carbonates. To avoid overloading of the article, these data are not included in the article.

Regarding the TEOS hydrolysis, we focused on the conditions, which allowed to obtain well-homogenized gel which easy to work with. Regarding the long-term stability in the atmosphere, we are not noticed any degradation of the material by repeated measuring the light output. Of course, a lot of study is ahead, what is typical for a new material. Quantitative estimation of various affecting factors including irradiation will be revealed in a separate article.

Point 2: The determination of lifetimes is not clear. Authors only stated that they used multiexponential function and 1 μs pulse width with 25 Hz repetition rate. I would be interesting how the nanosecond lifetimes (shorter than pulse width 1 μs) were extracted from decay data and what is the real accuracy of data fits (reduced Chi-square values) and lifetime values? Could authors provide instrument response function (pulse) and add it into data of Fig. 6? If the reconvolution has not been used, the obtained lifetimes are probably faulty since they are affected by pulse kinetics. Moreover, the decay data of Fig. 6 should be normalized between background (minimum) and maximum values, i.e. not only to maximum.

Response 2: Details of the techniques for evaluation of the luminescent and scintillation parameters are rearranged.

For the luminescence kinetics measurements description, we described a pulse width of the LED excitation, which is 500 ps. The excitation pulse also was included in Fig.6.

Point 3: Please, add the instrument response function (pulse) into Fig. 8.

Response 3: This information is included in line 277.

Response functions of PMT were measured to be 1.2+/-0.2 ns.

Point 4: In Fig. 7 should be added background of measurement to proof that the observed spectra correspond only to samples emission. What is the reason of large signal distortion at longer channels? Has been Compton scattering taken into account?

Response 4: In this study, we utilized alpha-particles for measurements. In pulse height spectra, they provide so-called full absorption peak. No Compton scattering effect is observed for charged particles. A distortion of the full absorption peak is provided by combining the fluctuation of the energy losses of the alpha-particles and, the dependence of the light extraction on the penetration depth when detecting light from a composite screen.

Typically, the background level is not included in the pulse height spectra. The counts from background and electronic noise can be estimated by consideration of the initial part of the spectra.

Point 5: I would be interested whether it is possible to compare pulse height spectra of samples of totally different geometry, i.e. single crystalline bulk YAG:Ce garnet to 200 μm thick composite (Li2CaSiO4:Eu/polyacrylate) film on Al substrate? For example, the different geometry studied in [M. Sasano, H. Nishioka, S. Okuyama et al., Nuclear Instruments and Methods in Physics Research A 715 (2013) 105] affects the resulting light yield. Therefore, the additional measurement of YAG:Ce prepared in same way like Li2CaSiO4:Eu samples, i.e. crushed and polyacrylate bonded to form a 200 μm thick film, is highly desired to be measured as a reference. It may affect the determination of the light yield by using the Eq. (3). Moreover, the sensitivity of detector to number of detected photons is not linear and detector saturation may be present at high-count rates. Did authors consider those facts in determination of light yield?

Response 5: We agree with the Reviewer that light yield measurement result is sensitive to geometry and light collection in the sample. In particular, it is related to the measurements with gamma-quanta when penetration depth is quite high even in dense materials. When alpha-particles are utilized, the small penetration depth diminish the volume effects. For instance, the penetration depth of 5 MeV alpha-particles in the Li2CaSiO4 and reference sample is ~6 μm by our GEANT4 simulation. Thus, we can neglect the difference between shapes of samples and a reference.

Chapter 3.5 updated (lines 260-262 and 265-270). New reference was included.

Point 6: Authors stated that the studied samples may be utilized for scintillation. However, the studied samples have too long decay times thus, those materials could be hardly utilized as scintillators where is desired lifetime in scale of <100 ns (due to presence of Eu3+ traces?), rather approaching the 1–10 ns. Nevertheless, the present study is important but the above mentioned points must be carefully clarified.

Response 6: We agree with the Reviewer that fast, high light yield and dense scintillator is very important for gamma-quanta registration. For neutrons, a combining of the high light yield and lightness of the material to avoid background registration is preferable. The fluxes of the neutrons in the state-of- art neutron measuring facilities are quite rarely exceeding 106 n/cm2. Therefore, the scintillation kinetics which is faster than 1 microsecond is a good progress in a comparison with widely used ZnS(Ag) (3 – 6 μs).

We expect good prospects for the developed material in neutron radiography (https://www.rctritec.com/en/scintillators/products/6-lif-based-scintillators.html), or for the simultaneous alpha-, beta- detection (https://eljentechnology.com/products/zinc-sulfide-coated/ej-444 ).

Reviewer 2 Report

The current work focuses on Eu-doped Li2CaSiO4 powder on scintillation properties. In the contents, I do not find any fatal errors, and I can recommend a publication after some modifications listed below.

1.       In the title or abstract, please comment that the material is powder. Before agreeing the review, judging from the title and abstract, I imagine a bulk crystal since uses of powder is quite limited in scintillator field (most scintillators are bulk crystal, ceramic or glass).

2.       For the assignment of Eu2+ emission in PL, please show the data of undoped one since undoped materials sometimes show similar broad emission with dopants.

3.       In Table 1, please comment origins of each component. In this case, Eu2+ and host (defects?) will be easy assignment, but what is the third component?

4.       In the calculation of light yield, you should consider about the quantum efficiency (QE) of PMT. When I check the catalogue, QE at 460 nm is ~20% and that at 520 nm is ~10%. This comment is also true for other results (Fig.9). Further, light yields calculated by alpha-ray base and actual experiment by neutron are necessarily the same. It should be commented.

5.       Why do you measure 511 keV gamma-ray induced scintillation decay? From the logic, neutron (or alpha-ray) induced ones should be measured. In addition, you should compare scintillation decay with PL ones.

6.       Regarding to Fig. 9, do you have a significant meaning for “13460 vs. 11520 ph./MeV”? It is common that this kind of measurement has a systematic error of +/-10% by optical attachment. Possible conclusion from this experiment is that the impurity does not affect the pulse height.

7.       After reading, to my impression, more discussion is required. Except for 2.4, the contents are something like a technical report.

Author Response

Dear Reviewer,

Thank you for comments, they will certainly will help to improve the article.

Please see the attachment our revised manuscript. All corrections were made with MS Word track changes function. 

Below please see our response to your comments.

Point 1: In the title or abstract, please comment that the material is powder. Before agreeing the review, judging from the title and abstract, I imagine a bulk crystal since uses of powder is quite limited in scintillator field (most scintillators are bulk crystal, ceramic or glass).

 Response 1: We agree with the Reviewer. Mentioning about the powder material in the abstract will be useful. Abstract was modified accordingly (please see lines 14-15 and 19).

Point 2: For the assignment of Eu2+ emission in PL, please show the data of undoped one since undoped materials sometimes show similar broad emission with dopants

Response 2: We added PL spectra for the undoped sample. A new curve was added to Fig 4.

Point 3: In Table 1, please comment origins of each component. In this case, Eu2+ and host (defects?) will be easy assignment, but what is the third component?

Response 3: Appropriate sentences are included in the text, lines 148-154.

At the lowest concentration of activator, the kinetics is close to single exponential with a decay time of a half microsecond. Further increases in the doping concentration provide an acceleration of the initial part of the kinetics due to concentration quenching. It results in the appearance, at approximation, the shorter components. At a multicomponent approximation of the luminescence kinetics, the drop in the luminescence yield can be characterized by a change in the effective decay constant.

Point 4: In the calculation of light yield, you should consider about the quantum efficiency (QE) of PMT. When I check the catalogue, QE at 460 nm is ~20% and that at 520 nm is ~10%. This comment is also true for other results (Fig.9).

Further, light yields calculated by alpha-ray base and actual experiment by neutron are necessarily the same. It should be commented.

Response 4: In the section Materials and Methods section (3.5), we mentioned about the difference in the quantum efficiency of the PMT photocathode at the YAG:Ce and Li2CaSiO4:Eu emission wavelengths. So horizontal scales in Figs 7 and 9 were adjusted to the same QE of PMT for Li2CaSiO4:Eu2+ (16% at λmax = 480 nm) and YAG:Ce (9% at λmax = 540 nm).

As follows from Eq.1, the alpha particle is a product of the 6Li interaction with the neutron. For this reason, future excitation of the light yield can be provided with alpha-particles.

Point 5 Why do you measure 511 keV gamma-ray induced scintillation decay? From the logic, neutron (or alpha-ray) induced ones should be measured. In addition, you should compare scintillation decay with PL ones.

Response 5: Measurement of the scintillation kinetics with the start-stop method with annihilation gamma-quanta is a well-recognized technique. The digitizing of the scintillation pulse under alpha-particles with so- called digitizers, like DRS-4, requires special fast electronics. Available neutron sources provide low- rate fluxes, so acquisition of reliable statistics would be enormously long.

Point 6. Regarding to Fig. 9, do you have a significant meaning for “13460 vs. 11520 ph./MeV”? It is common that this kind of measurement has a systematic error of +/-10% by optical attachment. Possible conclusion from this experiment is that the impurity does not affect the pulse height.

 Response 6: We agree with the Reviewer, the error of the measurements is included in the revised version. The text (lines 217-220 are modified accordingly).

Reviewer 3 Report

The authors submitted the article about the novel light silicate scintillator doped with Eu2+. The results are interesting and promising, the motivation for the study is clear and the experimental data is relevant. The conclusions are correct. There are few insufficiencies which require a minor revision.

1. The authors draw the conclusion about the transparency of the material based solely on the microscope image. I believe this point would be stronger if authors provided a transmittance spectrum of the prepared coating or at least a photograph of the coating.

2.  The caption of Fig. 5b seems incorrect. This is not a ratio between two logs, just log(I/x) in function of log(x). Also the function of the fitting curve is unclear and the conclusion about the concentration-dependent quenching (as opposed to (?)-dependent quenching?)

3. The conclusion on the decay curves could be more detailed – e.g. comparison with other scintillator materials, mechanisms of luminescence dynamics etc.

4. Fig. 7 inset is too small.

5. The should be also light yield comparison to other scintillating materials to evaluate the performance of this proposed material.

Author Response

Dear Reviewer, 

Thank you for the comments! I hope that after corrections our article will be more attractive to readers.

Please see the attachment our revised manuscript. All corrections were made with MS Word track changes function. 

Below please see our response to your comments.

Point 1: The authors draw the conclusion about the transparency of the material based solely on the microscope image. I believe this point would be stronger if authors provided a transmittance spectrum of the prepared coating or at least a photograph of the coating.

 Response 1: We agree with the Reviewer. A new panel with a photograph of powder layer is included in Fig.1

Point 2: The caption of Fig. 5b seems incorrect. This is not a ratio between two logs, just log(I/x) in function of log(x). Also the function of the fitting curve is unclear and the conclusion about the concentration-dependent quenching (as opposed to (?)-dependent quenching?)

Response 2: We agree with the Reviewer. The panel b, which presents data in logarithmic scales, is a specific presentation for technologists. To avoid discussion, we removed this plot from the revised article.

Point 3: The conclusion on the decay curves could be more detailed – e.g. comparison with other scintillator materials, mechanisms of luminescence dynamics etc.

Response 3: A new reference and appropriate sentences are. included.

Point 4: Fig. 7 inset is too small.

Response 4: Fig. 7 was updated. Also, we supply al the images in high resolution in zip-archive.

Point 5: The should be also light yield comparison to other scintillating materials to evaluate the performance of this proposed material.

Response 5: We added this information to the conclusions. 3 new referenced were added.

Round 2

Reviewer 1 Report

The manuscript has been improved over its original. Authors satisfactorily explained all ambiguities. Therefore, I recommend the manuscript for publication.

I am only recommending to read carefully the manuscript and correct the decimal separators in numbers which should be dot instead of comma (e.g. Figures 2-8, Tables 1,4,5). Only from this reason, my recommendation is "minor revision" relating to text editing. The scientific content is OK.

Reviewer 2 Report

Now, the paper is worth for publication.